# Evaluation of Immunodiagnostic Performances of *Neospora caninum* Peroxiredoxin 2 (NcPrx2), Microneme 4 (NcMIC4), and Surface Antigen 1 (NcSAG1) Recombinant Proteins for Bovine Neosporosis

**DOI:** 10.3390/ani14040531

**Published:** 2024-02-06

**Authors:** Ruenruetai Udonsom, Poom Adisakwattana, Supaluk Popruk, Onrapak Reamtong, Charoonluk Jirapattharasate, Tipparat Thiangtrongjit, Sarinya Rerkyusuke, Aran Chanlun, Tanjila Hasan, Manas Kotepui, Sukhontha Siri, Yoshifumi Nishikawa, Aongart Mahittikorn

**Affiliations:** 1Department of Protozoology, Faculty of Tropical Medicine, Mahidol University, Bangkok 10400, Thailand; ruenruetai.udo@mahidol.ac.th (R.U.); supaluk.pop@mahidol.ac.th (S.P.); 2Department of Helminthology, Faculty of Tropical Medicine, Mahidol University, Bangkok 10400, Thailand; poom.adi@mahidol.ac.th; 3Department of Molecular Tropical Medicine and Genetics, Faculty of Tropical Medicine, Mahidol University, Bangkok 10400, Thailand; onrapak.rea@mahidol.ac.th (O.R.); tipparat.thi@mahidol.ac.th (T.T.); 4Department of Pre-Clinic and Animal Science, Faculty of Veterinary Science, Mahidol University, Nakhon Pathom 73170, Thailand; charoonluk.jir@mahidol.edu; 5Division of Livestock Medicine, Faculty of Veterinary Medicine, Khon Kaen University, Khon Kaen 40002, Thailand; sarinyare@kku.ac.th (S.R.); aran_jan@kku.ac.th (A.C.); 6Department of Medicine and Surgery, Faculty of Veterinary Medicine, Chattogram Veterinary and Animal Sciences University, Chattogram 4225, Bangladesh; tanjila@cvasu.ac.bd; 7National Research Center for Protozoan Diseases, Obihiro University of Agriculture and Veterinary Medicine, Obihiro 080-8555, Japan; 8Medical Technology, School of Allied Health Sciences, Walailak University, Nakhon Si Thammarat 80160, Thailand; manas.ko@wu.ac.th; 9Department of Epidemiology, Faculty of Public Health, Mahidol University, Bangkok 10400, Thailand; sukhontha.sir@mahidol.ac.th

**Keywords:** *Neospora caninum*, neosporosis, cattle, bovine, diagnosis, recombinant proteins

## Abstract

**Simple Summary:**

Bovine neosporosis is a widespread and economically impactful disease affecting the dairy and meat industries globally. With no effective drug or vaccine available, the disease’s control relies on accurate detection methods. This study assessed the performance of three *Neospora caninum* proteins—NcPrx2, NcMIC4, and NcSAG1—as immunodiagnostic tools for identifying IgG antibodies against *N. caninum*. Comparisons were made with the indirect fluorescent antibody test (IFAT). The findings indicate that NcSAG1 exhibited the highest sensitivity and specificity, followed by NcMIC4. In contrast, NcPrx2 demonstrated lower sensitivity and specificity compared with IFAT. This study highlights that NcSAG1 is helpful as antigen marker and that NcPrx2 and NcMIC4 have potential in immunodiagnosis for detecting *N. caninum* infections in field samples. These findings could contribute to improved treatment management, surveillance, and risk assessment of neosporosis in livestock in the future.

**Abstract:**

Bovine neosporosis is among the main causes of abortion in cattle worldwide, causing serious economic losses in the beef and dairy industries. A highly sensitive and specific diagnostic method for the assessment of the epidemiology of the disease, as well as it surveillance and management, is imperative, due to the absence of an effective treatment or vaccine against neosporosis. In the present study, the immunodiagnostic performance of *Neospora caninum* peroxiredoxin 2 (NcPrx2), microneme 4 (NcMIC4), and surface antigen 1 (NcSAG1) to detect IgG antibodies against *N. caninum* in cattle were evaluated and compared with that of the indirect fluorescent antibody test (IFAT). The results revealed that NcSAG1 had the highest sensitivity and specificity, with values of 88.4% and 80.7%, respectively, followed by NcPrx2, with a high sensitivity of 87.0% but a low specificity of 67.0%, whereas NcMIC4 showed sensitivity and specificity of 84.1% and 78.9%, respectively, when compared with IFAT. A high degree of agreement was observed for NcSAG1 (k = 0.713) recombinant protein, showing the highest diagnostic capability, followed by NcMIC4 (k = 0.64) and NcPrx2 (k = 0.558). The present study demonstrates that NcSAG1 is helpful as an antigen marker and also demonstrates the potential immunodiagnostic capabilities of NcPrx2 and NcMIC4, which could serve as alternative diagnostic markers for detecting *N. caninum* infection in cattle. These markers may find utility in future treatment management, surveillance, and risk assessment of neosporosis in livestock or other animal host species. Further research should be directed toward understanding the in vivo immune response differences resulting from immunization with both recombinant proteins.

## 1. Introduction

Neosporosis, caused by *Neospora caninum*, is an economically important disease in cattle. This infectious disease is associated with various reproductive complications, such as abortion, stillbirth, and the delivery of weakened calves [1,2]. The prevalence of *N. caninum* is widespread on a global scale, making it one of the predominant causes of bovine abortion and resulting in economic losses in both beef and dairy industries worldwide [3]. Vertical transmission is the most important route of parasite transmission and plays a pivotal role in sustaining the infection within the cattle population [4,5]. Neosporosis often presents as asymptomatic, although it can also manifest symptoms or cause abortions. However, the efficiency of transplacental transmission of *N. caninum* in cattle has been estimated to range from 44 to >95% [6,7], and the percentage of abortions is higher in seropositive dams than in seronegative dams [8,9,10]. To date, there is no effective chemotherapeutic treatment or vaccine for the prevention of neosporosis. Consequently, the management of the disease hinges on the utilization of accurate detection tests, influencing the formulation of control strategies [11].

Several serological methods, including the indirect fluorescent antibody test (IFAT), enzyme-linked immunosorbent assays (ELISAs), immunoblot, and direct agglutination tests, are available, and these tests are used to detect specific antibodies against *N. caninum* in cattle [12]. Among these assays, IFAT is considered the reference technique for detecting antibodies to *N. caninum* [13]. However, using intact tachyzoites as antigens in the IFAT method may detect cross-reactivity with antibodies against other members of the phylum Apicomplexa, including the protozoan *Toxoplasma gondii* [14]. Besides IFAT, another effective diagnostic method utilized for detecting *N. caninum* infections in a large number of animals is the indirect ELISA (iELISA), which employs recombinant proteins from *N. caninum*. This approach has been reported to have considerable sensitivity and excellent diagnostic accuracy [15,16].

Immunoproteomics is a powerful tool for identifying potential immunogenic antigens against *N. caninum* infection [17,18,19]. In a previous study, *N. caninum* species-specific antigens were identified using bovine infected sera through immunoproteomic analysis. Based on two-dimensional electrophoresis immunoblotting, 14 different antigenic proteins were specific to *N. caninum*. Among these, peroxiredoxin 2 (Prx2) and microneme 4 (MIC4) exhibited high immunoreactivity specificity against *N. caninum* infection [20]. In the pursuit of a reliable diagnostic method and vaccine development, various *N. caninum* recombinant proteins have been investigated as potential target antigens and vaccine candidates. 

*N. caninum* peroxiredoxin (NcPrx) was recently reported to have peroxidase and antioxidant functions. The recombinant NcPrx (rNcPrx) protein exhibited a strong reaction with the polyclonal anti-rNcPrx serum, as observed with native NcPrx [21]. However, the antigenicity of NcPrx remains unknown. In *T. gondii*, recombinant proteins TgPrx1 and TgPrx3 induced protection against *T. gondii*-infected mice and were evaluated as potential vaccine candidates against toxoplasmosis [22,23]. Additionally, *Leishmania* peroxidoxin 1 (LdPxn1) elicited a robust CD4+ T cell response, resulting in partial protection against cutaneous leishmaniasis in immunized mice [24].

*N. caninum* microneme 4 (NcMIC4) is an associated protein found within the micronemes, and it is released by the parasite as a soluble component during host cell entry [25]. Experiments involving recombinant NcMIC4 have demonstrated a significant increase in protective immunity against neosporosis in a mouse model [26]. For *T. gondii*, immunization of mice with TgMIC1 and TgMIC4 triggers a protective immune response against *T. gondii* infections [27]. Additionally, a combination of TgMIC1/TgMIC4/TgMIC6 recombinant proteins significantly heightens an effective immune response, leading to a reduction in the mortality rate of mice [28]. 

The surface antigen 1 (NcSAG1) has been recognized as one of the most immunogenic and efficacious antigens for diagnosing bovine neosporosis [29]. Recently, the NcSAG1-based iELISA was developed, demonstrating remarkable antigenic properties and facilitating the assessment of *Neospora*-induced abortions in cattle [30]. Moreover, the utilization of NcSAG1 in an immunochromatographic assay (ICT) was consistently effective in detecting anti-*N. caninum* antibodies in cattle field samples [31]. 

The number of *N. caninum* proteins that have been investigated as diagnostic targets is limited. Therefore, the present study aimed to comprehensively assess the following three *N. caninum* recombinant antigens: NcPrx2, NcMIC4, and NcSAG1. Our study objective was to evaluate their serodiagnostic performance and compare their efficacy with that of IFAT, an established reference method for diagnosing neosporosis in cattle.

## 2. Materials and Methods

### 2.1. Preparation of N. caninum Tachyzoites

African green monkey kidney (Vero) cells were cultured in Dulbecco’s modified Eagle medium (Cytiva HyClone™, South Logan, UT, USA), supplemented with 8% fetal bovine serum (FBS), L-glutamine (2 mM/mL), penicillin–streptomycin (100 U/mL of penicillin and 100 μg/mL of streptomycin), and amphotericin B (0.25 µg/mL), in a humidified atmosphere with 5% CO_2_ at 37 °C until a confluent monolayer of cells was achieved. Subsequently, the cells were transferred to a maintenance medium containing 2% FBS and inoculated with *N. caninum* (Nc-1) tachyzoites. The culture was then incubated at 37 °C with 5% CO_2_ for 24 h. A serum-free cell culture medium (without FBS) was introduced and maintained until the tachyzoites were harvested at approximately 48 h. The purification of tachyzoites involved passing them through a 27-gauge needle and a 5 µm filter. The parasites were washed twice with phosphate-buffered saline (PBS), counted, and then utilized for the IFAT.

### 2.2. Production of GST-Fused Recombinant Proteins NcPrx2, NcMIC4, and NcSAG1

The recombinant NcSAG1 protein was expressed and purified following previously described protocols [30]. Similar to NcSAG1, recombinant NcPrx2 and NcMIC4 proteins were expressed in *Escherichia coli* BL21 (DE3) under optimal conditions. The target sequences were polymerase chain reaction (PCR)-amplified using specific primers containing suitable restriction enzyme sites, and the resulting proteins were expressed as glutathione S-transferase (GST) fusions (New England BioLabs Inc., Ipswich, MA, USA). Briefly, the PCR tests were performed with *N. caninum* cDNA (Nc-1 strain) as the template. The truncated lengths of the NcPrx2 (NCLIV_053640) and NcMIC4 (NCLIV_002940) genes were generated through PCR using specific primers, which were as follows: NcPrx2_Forward, 5′-TGGATCCCCGGAATTAATGAGTCACCCCCATGAT-3′, and reverse, 5′-GATGCGGCCGCTCGACTAAGCCGAAGGATCTGG-3′; and NcMIC4_ Forward, 5′-TGGATCCCCGGAATTAATGACTATAGGTGGTGACG-3′, and reverse, 5′-GATGCGGCCGCTCGATTATGCGTCTTCCTCTTCAA-3′; respectively. The PCR products were purified from agarose gels and cloned into the pGEX-4T-1 expression vector, which was treated with *EcoR*1 and *Xho*1 restriction enzymes. Successful insertion was confirmed by DNA sequencing. The resulting recombinant plasmids were introduced into *E. coli* BL21 (DE3) cells for protein expression. Bacterial cells that were transformed with NcPrx2 and NcMIC4 recombinant plasmids were cultured in liquid LB media supplemented with 50 µg/mL ampicillin and incubated at 37 °C for 14 h. The bacterial cultures were then transferred to fresh liquid LB media and incubated until an optical density of 0.5–0.7 was reached. NcPrx2 and NcMIC4 were induced in the culture with 0.1 mM isopropyl β-D-1-thiogalactopyranoside and incubated at 37 °C for 2 and 4 h, respectively. The bacterial cells were harvested, then centrifuged at 7180× *g* at 4 °C for 20 min. The pellets were suspended in a sonication buffer [50 mM Tris-HCl, pH 8, 50 mM NaCl, 1 mM ethylenediaminetetraacetic acid (EDTA), and 1 mM dithiothreitol] along with lysozyme with a final concentration of 500 µg/mL and incubated on ice for 30 min. The suspension was sonicated and Triton X-100 (10%) in PBS was added, followed by incubation on ice for 20 min and subsequent centrifugation at 7180× *g* at 4 °C for 20 min. The resulting supernatant was applied to Glutathione Sepharose 4B beads (GE Healthcare Life Sciences, Buckinghamshire, UK) as per the manufacturer’s instructions. Specifically, the supernatant was incubated with washed beads at 4 °C overnight with gentle rotation. GST fusion proteins were eluted with an elution buffer (100 mM Tris-HCl, pH 8.0, containing 100 mM NaCl, 5 mM EDTA, and 25 mM reduced glutathione powder; MP Biomedicals GmbH., Eschwege, Germany) and incubated at 4 °C overnight with mild rotation before centrifugation at 800× *g*, 4 °C for 3 min. The purified recombinant proteins were analyzed through Sodium Dodecyl Sulphate-Polyacrylamide Gel Electrophoresis (SDS-PAGE) and stained with Coomassie Brilliant Blue R250 (MP Biomedicals Inc., Illkirch-Graffenstaden, France). Protein concentrations were quantified using a bicinchoninic acid protein assay kit (Thermo Fisher Scientific, Inc., Rockford, IL, USA). Finally, NcPrx2, NcMIC4, and NcSAG1 recombinant proteins were identified and confirmed via liquid chromatography-mass spectrometry (LC-MS/MS).

### 2.3. Protein Identification by Mass Spectrometry

Each protein band section was excised from 1-D SDS-PAGE and de-stained overnight at 4 °C with 50% acetonitrile (ACN; Sigma-Aldrich, St. Louis, MO, USA) in 50 mM ammonium bicarbonate (Merck, Rahway, NJ, USA). The protein samples were reduced using 4 mM Dithiothreitol (DTT) in 50 mM ammonium bicarbonate, and this reduction process occurred at a temperature of 60 °C for 15 min. Following reduction, the proteins were alkylated with 250 mM iodoacetamide at room temperature for 30 min in the absence of light. The reaction was quenched using 4 mM DTT in 50 mM ammonium bicarbonate for 5 min. Subsequently, all solutions were removed, and the gel band was dehydrated using acetonitrile. The protein samples were digested with trypsin proteomics grade (Sigma-Aldrich, St. Louis, MO, USA) overnight at 37 °C in 50 mM ammonium bicarbonate. The resulting digested peptides were extracted using acetonitrile and then dried in a vacuum centrifuge. The dried tryptic peptides were analyzed using an UltiMate 3000 nano-liquid chromatography (nano-LC) system (Dionex, Surrey, UK). The mass spectra from both mass spectrometry (MS) and tandem mass spectrometry (MS/MS) covered mass ranges of *m*/*z* 400–2000 and *m*/*z* 50–1500, respectively. A mascot generic file (.mgf) was generated using data analysis 4.1 software (Bruker Daltonics, Billerica, MA, USA). For merging the .mgf files and protein identification, Mascot Daemon version 2.3.2 (Matrix Science, London, UK) was employed. Protein identification was executed using the Mascot algorithm search on the National Center for Biotechnology Information database (https://www.ncbi.nlm.nih.gov/) (accessed on 9 June 2023) and *Toxoplasma* informatics resource (http://www.ToxoDB.org) (accessed on 9 June 2023) protein databases.

### 2.4. Serum Samples

#### 2.4.1. Control Sera

NcPrx2, NcMIC4, and NcSAG1 recombinant proteins were assessed using sera from mice experimentally infected with parasites and sera from infected bovines. The infected mouse sera were obtained from female ICR mice aged 6–8 weeks provided by the National Laboratory Animal Center, Mahidol University, Salaya Campus, Thailand. The mice were housed at the Laboratory Animal Science Unit, Faculty of Tropical Medicine, Mahidol University. Intraperitoneal inoculation of 1 × 10^5^
*N. caninum* or 1 × 10^3^
*T. gondii* tachyzoites was performed [30]. Blood samples were collected from the mice on day 0 and at 8 weeks post-inoculation. The infected and uninfected bovine sera, confirmed by IFAT and iELISA against NcSAG1, were provided by the National Research Center for Protozoan Diseases, Obihiro University of Agriculture, and Veterinary Medicine, Obihiro, Hokkaido, Japan [32]. Bovine sera infected with other protozoan parasites, including *T. gondii* (2 samples), *C. parvum* (2 samples), *Babesia bovis* (2 samples), and *Babesia bigemina* (2 samples), which were confirmed by at least two serology tests, were also included for specificity testing [20].

#### 2.4.2. Field Dairy Cattle Serum Samples

Altogether, 247 dairy cattle serum samples from 8 cattle farms were collected from randomly selected herds from smallholder dairy farmers in Khon Kaen Province, Thailand. Khon Kaen is a city located in the northeastern region of Thailand and is one of the four major cities in the region. It lies approximately 450 km northeast of Bangkok (Figure 1). According to the records from Thailand’s Regional Livestock Office in 2021, Khon Kaen Province has 38,652 dairy cattle. The majority of cattle in this area are raised for milk production (Regional Livestock Office 3, Thailand, 2021). The cattle were confirmed to be clinically healthy and belonged to the Holstein-Friesian cross-breed. These cattle were uniformly accommodated in conventional open structures, covering a wide age range from 1 to 14 years. Their dietary regimen consisted of a combination of roughage and concentrate, with grazing opportunities allowed during the rainy season. Remarkably, no cases of abortion were recorded by the farmers in association with this cattle cohort. It is noteworthy that dogs were consistently present on all farms, and neighboring dogs had the potential to interact with the cattle. Blood samples were collected from either the jugular or caudal vein. These samples were promptly transferred into 10 mL vacuum blood tubes. All samples were placed in a styrofoam box, kept cool with ice packs, and transported to the laboratory. Subsequently, the sera were separated and stored at −20 °C until examination. Our study experiments involving mice and field cattle samples were approved by the Animal Care and Use Committee of the Faculty of Tropical Medicine-Animal Care and Use Committee (FTM-ACUC), Mahidol University, Bangkok, Thailand (Approval No. FTM-ACUC 027/2020).

### 2.5. Indirect ELISA (iELISA)

In our study, 96-well microtiter plates (Nunc, Roskilde, Denmark) were coated with purified recombinant protein at 4 °C overnight. The final concentrations were 0.1 μM for NcMIC4 and NcSAG1, and 0.5 μM for NcPrx2, in a 50 mM carbonate-bicarbonate buffer (pH 9.6). The plates were washed once with PBS containing 0.05% Tween 20 (PBS-T) and then blocked with 3% skimmed milk in PBS (PBS-SM) for 1 h at 37 °C. Following the blocking step, the plates were washed again with PBS-T, and 50 μL of the test sera (each sample in duplicate) diluted at 1:200 was added to the wells. The plates were incubated at 37 °C for 1 h. After washing, a secondary antibody, peroxidase-conjugated anti-mouse or bovine IgG (Invitrogen, Rockford, IL, USA), diluted at 1:4000 in PBS-SM, was added to the wells and incubated at 37 °C for 1 h. The plates were washed six times, and then 100 μL of 3,3′, 5,5′-tetramethylbenzidine substrate (Thermo Fisher Scientific, Frederick, MD, USA) was added to each well. The reaction was stopped by adding 100 μL of 1-M HCl. The absorbance at 450 nm was measured using the SUNRISE microplate reader (Tecan Austria GmbH, Grödig, Austria). The cut-off value was determined as the mean value of the standard *N. caninum*-negative control sera plus five times the standard deviation.

### 2.6. IFAT

Field cattle serum samples were evaluated for *N. caninum* infection using IFAT as the standard test and compared with iELISA utilizing NcPrx2, NcMIC4, and NcSAG1 recombinant proteins. The IFAT procedures were performed as described previously [33]. Briefly, killed *N. caninum* whole tachyzoites were fixed on printed microscope slides (Epredia™, Portsmouth, NH, USA). The serum samples were analyzed at a dilution of 1:200 [33,34]. The diluted sera were incubated on antigen-coated slides at 37 °C for 1 h in a moist chamber. Subsequently, the slides were washed with a rinse buffer (VMRD, Inc., Pullman, WA, USA). Fluorescein-labeled antibody to bovine IgG (SeraCare, Milford, MA, USA) was added and incubated in a moist chamber for 1 h. After washing, the slides were mounted in buffered glycerol and examined under a fluorescent microscope (ZEISS Axio Imager M2, Göttingen, Germany). Positive samples were identified by the presence of complete peripheral fluorescein around the *N. caninum* tachyzoites, whereas a negative result was indicated by the absence of fluorescence signal (Appendix A).

### 2.7. Statistical Analysis

The diagnostic performance of the iELISA assays, including the standard cut-off point, sensitivity, and specificity, was determined by receiver operating characteristic (ROC) analysis using PASW Statistics 18.0 (Chicago, IL, USA: SPSS Inc.; 2009). Test agreement (Kappa values; k) with 95% confidence intervals (95% CI) was calculated using VassarStats: Website for Statistical Computation (http://vassarstats.net) (accessed on 2 November 2023). The strength of agreement was categorized based on the kappa values as follows: fair (0.21–0.40), moderate (0.41–0.60), substantial (0.61–0.80), and almost perfect (0.81–1.00) [35].

## 3. Results

### 3.1. Identification of NcPrx2, NcMIC4, and NcSAG1 Recombinant Proteins

The analysis of NcPrx2, NcMIC4, and NcSAG1 recombinant proteins using 12% SDS-PAGE revealed estimated molecular weights of 47.2, 86.1, and 55 kDa, respectively (Appendix A). Additionally, through further identification by mass spectrometry, all the proteins exhibited correct amino acid sequences (Appendix A).

### 3.2. Assessment of Recombinant Proteins Using Mouse and Bovine Infected Sera

The diagnostic performance of NcPrx2, NcMIC4, and NcSAG1-based iELISA for *N. caninum* infection was assessed using sera from experimentally infected mice and cattle. The highest reactivity was detected with NcSAG1, followed by NcMIC4 and NcPrx2, in the sera obtained from mice experimentally infected with *N. caninum* (Appendix A). No cross-reactivity was observed in the sera from *T. gondii*-infected animals. When using the known bovine sera infected and uninfected with *N. caninum*, strong and high signal reactivity was observed for all recombinant antigens with *N. caninum*-infected sera, whereas low signal reactivity was observed in other protozoan-infected bovine sera (Appendix A).

### 3.3. Comparison of ELISAs Using NcPrx2, NcMIC4, and NcSAG1 with IFAT in Field Cattle Serum Samples

We evaluated the antigenic efficacy of three recombinant antigens in detecting *N. caninum* infection in cattle using iELISA and compared the results with those of the standard IFAT. Among the 247 tested sera, 62.7% of the samples were positive for NcPrx2, 55.8% for NcMIC4, 55.4% for NcSAG1, and 55.8% by IFAT (Table 1). 

Receiver Operating Characteristics (ROC) analysis was conducted to assess the performance of indirect ELISA with recombinant proteins and field cattle sera. The area under the curve (AUC) values for NcPrx2, NcMIC4, and NcSAG1 were 0.808, 0.790, and 0.854, respectively (Figure 2). Based on the ROC curve analysis, the optimal cut-off points were determined as 0.66 for NcPrx2 and NcMIC4, and 0.64 for NcSAG1. Samples were classified as positive when their values exceeded the respective cut-off points (Figure 3). Among these, NcSAG1 exhibited the highest sensitivity and specificity at 88.4% and 80.7%, respectively. NcPrx2 showed a high sensitivity of 87.0% but a lower specificity of 67.0%, while NcMIC4 displayed sensitivity and specificity of 84.1% and 78.9%, respectively (Table 2 and Table 3). Notably, substantial agreement was observed for NcSAG1 and NcMIC4, with kappa values of 0.713 and 0.64, respectively. In contrast, NcPrx2 demonstrated moderate agreement with a kappa value of 0.558 (Table 3).

## 4. Discussion

Numerous serological methods for detecting bovine neosporosis have been reported [12]. Nonetheless, enhancing diagnostic precision remains imperative to enhance diagnostic sensitivity and specificity, considering the potential for serologic cross-reactivity among infections caused by related protozoan parasites in cattle [36]. In the present study, we expressed, and assessed the performance of three recombinant antigens from *N. caninum* for diagnosing bovine neosporosis. The NcSAG1-based iELISA exhibited substantial agreement, demonstrating a commendable sensitivity of 88.4% and specificity of 80.7% when compared with the values of IFAT. For NcMIC4-based iELISA and IFAT, the sensitivity, and specificity were 84.1% and 78.9%, respectively. In contrast, the NcPrx2-based iELISA and IFAT demonstrated a high sensitivity but low specificity, with values of 87.0% and 67.0%, respectively. Notably, NcSAG1 proves to be an effective serodiagnostic tool for both bovine and canine neosporosis [37,38,39] Moreover, it has displayed high sensitivity and specificity in cattle and ovine hosts [40,41]. Furthermore, our study results are consistent with the findings of a previous study in which NcSAG1-based iELISA, and ICT displayed heightened sensitivity, specificity, and substantial agreement, indicating remarkable efficacy in detecting anti-*N. caninum* antibodies in field samples from Japan [31]. This consistency with our results reinforces the effectiveness of NcSAG1-based methods for antibody detection in Thailand.

NcMIC4 plays a crucial role in the initial stages of host cell adhesion and invasion [25]. A recent immunoproteomics study highlighted the highly specific immunogenic nature of NcMIC4, recognized by bovine-infected sera, suggesting its potential for diagnostic use and vaccine development [20]. However, while NcMIC4 exhibited high sensitivity, specificity, and substantial agreement in detecting *N. caninum* antibodies in field cattle, its efficacy is somewhat lower compared with NcSAG1 in the present study. Further research is required to explore the potential of NcMIC4 as a vaccine candidate against *N. caninum* infections.

Peroxidoxins or peroxiredoxins (Prxs) have been identified as a group of antioxidants present in both eukaryotes and prokaryotes [42]. However, despite the previous findings indicating significant immunoreactivity of NcPrx2 against *N. caninum*, suggesting its potential as a marker for immunodiagnostic development [20], its sensitivity and specificity for detecting *N. caninum* antibodies in cattle field samples are limited. 

In this study, the performance of recombinant protein-based iELISA was compared with a standard IFAT. IFAT has been accepted as reference method for *N. caninum* serology [4,43,44,45] and has been a valuable standard for the development of other serological techniques [12,46,47]. Hence, we adopted IFAT as the reference test in this study to assess the diagnostic performance of the iELISA utilizing recombinant proteins. The cut-off titer in IFAT varies among laboratories, ranging from 1:100 to 1:640 for adult cattle and from 1:16 to 1:80 for detecting *N. caninum* antibodies in fetal sera [46,48]. A recommended cut-off value in IFAT for detecting *N. caninum* antibodies in adult cattle is 1:200 dilution [49,50], while lower cut-off values of 1:16–1:25 have been suggested for fetal fluids [48]. Additionally, using a high dilution cut-off titer in IFAT is considered appropriate to minimize cross-reactivity with related parasites, such as *T. gondii*, in serum samples [51,52]. Despite its widespread use for the detection of specific anti-*N. caninum* antibodies in cattle, IFAT is a labor-intensive technique, unsuitable for large-scale investigations, and its interpretation can be subjective [16,53]. The immunoblot has demonstrated high sensitivity and specificity for detecting *N. caninum* antibodies in cattle [53,54,55]. However, it is only performed in a limited number of laboratories as a routine tool for screening cattle sera due to its labor-intensive nature, specialized equipment requirements, and time-consuming process [53,56].

In contrast to an earlier study conducted in the same region, the overall prevalence of *N. caninum* infection was approximately four times higher (13% compared with >55% in this study) [57]. The differences in the prevalence pattern may be due to differences in the *N. caninum* purified recombinant antigen-based indirect ELISA used in this study, whereas the previous study was based on an ELISA utilizing parasite proteins incorporated into immunostimulating complexes as antigens. This substantial increase in prevalence underscores the widespread distribution of *N. caninum* infection within the dairy cattle population across the study areas. Furthermore, the prevalence of *N. caninum* in the present study surpassed values reported in other regions, such as 46.9% in dairy cattle from the northern area [58], 15% in dairy cattle, and 20.8% in beef cattle in the western Provinces of Thailand [39,59]. The increased prevalence observed in the current study compared with the earlier study could be attributed to the disparities in sampling locations, climatic conditions, study design, duration, and the utilization of diverse diagnostic tests or varying cut-off values [60]. In serological tests, various factors can lead to result variability, including differences in serological assays, titer and absorbance values, antigen composition and concentration, cut-off points and conjugated characteristics, as well as the sensitivity and specificity of the tests. This variability can also extend to the agreement between individual tests [4,12,46,48,50]. In addition, low levels of *N. caninum* antibodies in the serum can contribute to variations in results, both between different tests and different laboratories using the same diagnostic test [49]. Previous studies have described the variability in results when different antigens are used in serological tests, affecting the detection of antibodies directed at various epitopes of *N. caninum* [49,61]. The variation in test results in this study likely stems from the use of different antigen preparations or compositions. For instance, IFAT is prepared from whole *N. caninum* tachyzoites, while indirect ELISA employs recombinant proteins that carry distinct immunodominant epitopes. Additionally, the random sampling strategy may influence the differences in results observed in this study. As revealed by kappa statistical analysis, NcSAG1-iELISA, when compared with IFAT, demonstrated the highest level of agreement. This finding is valuable for the detection and diagnosis of *N. caninum* in cattle.

Currently, there is no available *N. caninum* vaccine or treatment for cattle or dogs. Control options for *N. caninum* infection in cattle farms have been modeled to reduce the infection [62]. Unfortunately, data collection was performed during the COVID-19 pandemic, which restricted authorized veterinarians’ farm access solely for animal health monitoring. Consequently, comprehensive information concerning the risk factors associated with *N. caninum* infection remained unavailable for this study. As evidenced by previous research reported in 2007, vertical transmission emerged as the primary route of *N. caninum* infection in this particular region [57]. Our study suggests that horizontal infection may play a major role in *N. caninum* infection in cattle in the studied area as the cattle were healthy and no cases of abortion were recorded by the farmers. This is also supported by the finding that seropositivity tends to increase with the age of cattle and the presence of dogs on all farms. At a practical level in this situation, domesticated and neighboring dogs play an important role, potentially increasing cattle exposure to *N. caninum* through contamination of feed with excreted oocysts in their feces [12]. The presence and number of dogs were found to be correlated with a high prevalence of *N. caninum* antibodies in dairy cattle [63,64]. A study by Inpankaew et al. (2014) reported that the high seroprevalence of *N. caninum* in dairy cattle was associated with the presence of farm dogs in herds in the northern provinces of Thailand [58]. In the current study, dogs were close to all the cattle farms, and consequently, the risk of exposure to the infection was increased. Therefore, reducing horizontal transmission of *N. caninum* in dairy herds is a useful goal in controlling the infection. To achieve this, preventing stray or neighboring dogs from entering the farm area, as well as protecting feedstuffs and drinking water from contamination with canine feces, is crucial [65]. Additionally, feeding dogs with commercial food and keeping them away from cattle are effective measures to decrease *N. caninum* infection in dairy herds. Further study is needed to investigate the relationship between the presence of dogs on dairy farms and the prevalence of *N. caninum* infection in Thailand. 

On the other hand, the age of the animal is demonstrated to be relatively significant for *N. caninum* infection. Guimarães et al. (2004) reported that cows aged 24 months or over showed an increased risk of *N. caninum* infection [66]. Other reports also indicate that older cattle exhibit higher seropositivity to *N. caninum*, suggesting a greater possibility of horizontal transmission of the infection [67,68,69]. In this study, where cattle ranged in age from 1 to 14 years, the aforementioned findings suggest that the increasing age of cattle is also a significant factor for *N. caninum* infection. Although vertical transmission is the principal route of infection in cattle [70,71] and cannot be ruled out, recent research demonstrated a high prevalence of *N. caninum* in bovine placenta, indicating vertical transmission as the main route for *N. caninum* infection in Phayao Province, Northern Thailand [72]. A profitable control strategy to reduce infection is to avoid retaining or breeding heifers from seropositive cows and to inseminate all seropositive dams using beef semen [65]. Culling seropositive animals from the herd is also useful in reducing the infection rate [73].

The ROC curve analysis reveals good performance by NcSAG1, followed in decreasing order by NcMIC4 and NcPrx2, in detecting *N. caninum*-specific IgG in cattle sera. However, NcMIC4 and NcPrx2, while showing high sensitivity, exhibit lower specificity, which may make them less feasible for screening purposes. Nevertheless, NcMIC4 demonstrates a substantial level of agreement, and this study provides preliminary insight into the diagnostic efficacy of NcMIC4 and NcPrx2 for *N. caninum* antibody detection. Further studies are needed to explore the potential immunodiagnostic performance of both antigens in other animal host species for *N. caninum* infection. Consequently, a more comprehensive examination is warranted to assess the in vivo immune responses induced by immunizing mice with both recombinant proteins. Moreover, a comparison between the recombinant protein-based iELISA and the standard IFAT, immunoblot or commercial serological diagnostic kits in cattle or other animal host species should be conducted concurrently using substantial sample sizes. This will contribute to a comprehensive evaluation of the diagnostic performance and validity of these markers in a larger context.

## 5. Conclusions

Notably, NcSAG1 demonstrated the highest diagnostic potential, and the results highlight its superior effectiveness for diagnosing neosporosis in cattle compared with IFAT. Our study offers preliminary insights into the antigenicity of NcPrx2 and NcMIC4, which showed high sensitivity but low specificity compared with NcSAG1. Therefore, NcSAG1 is helpful as an antigen marker for detecting antibodies to *N. caninum* in cattle. However, NcPrx2 and NcMIC4 could serve as alternative candidates. Furthermore, our study data revealed a significant prevalence of *N. caninum* infection among dairy cattle in the studied regions.

## Figures and Tables

**Figure 1 animals-14-00531-f001:**
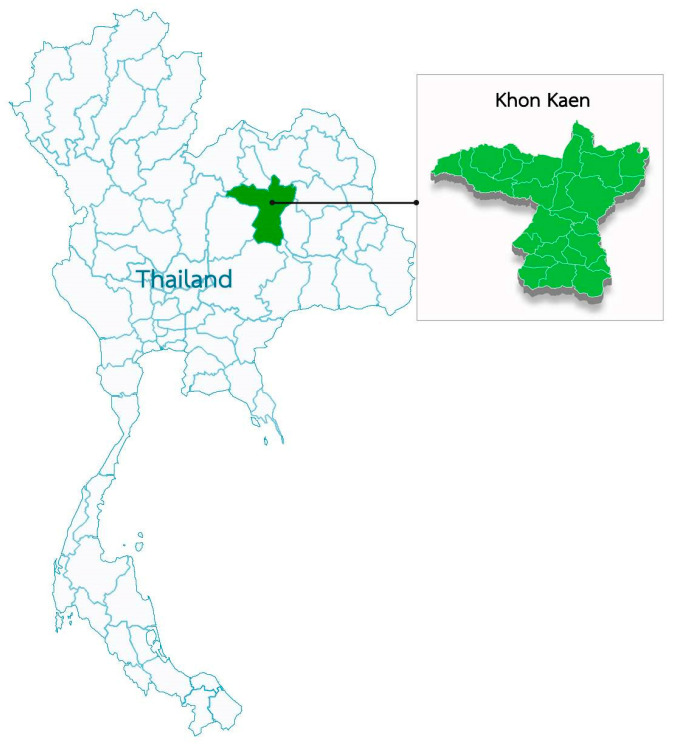
A map indicating the sampling sites within Khon Kaen Province, Thailand. The marker indicates the specific area under investigation in this study.

**Figure 2 animals-14-00531-f002:**
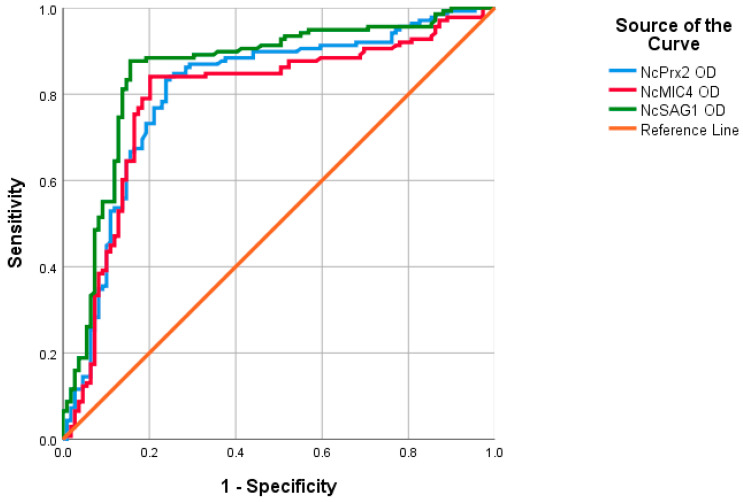
Receiver Operating Characteristics (ROC) analysis for the indirect ELISA with NcPrx2, NcMIC4 and NcSAG1 proteins using field cattle sera.

**Figure 3 animals-14-00531-f003:**
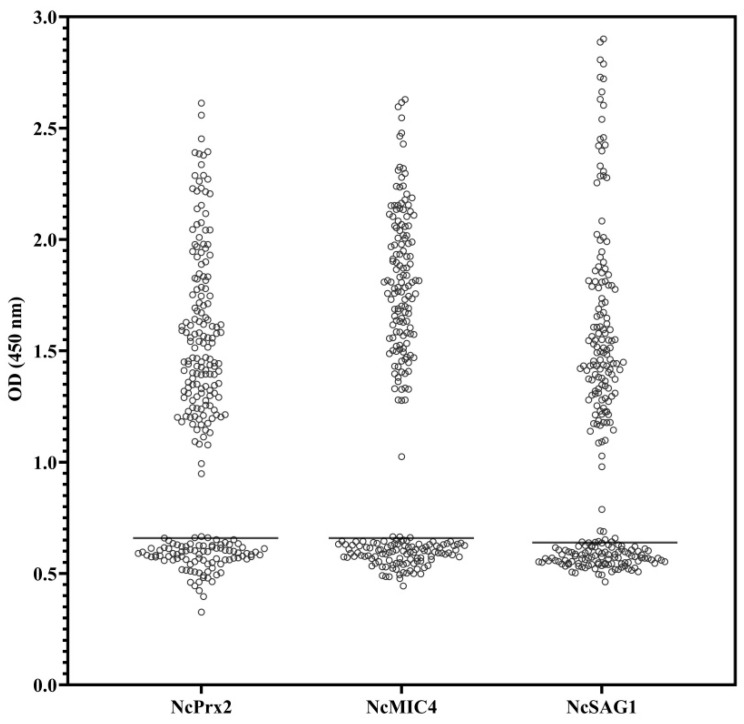
Evaluation of *N. caninum*-derived peroxiredoxin 2 (NcPrx2), microneme 4 (NcMIC4), and surface antigen 1 (NcSAG1) recombinant antigens using field cattle serum samples (n = 247), as an optimal cut-off determined by the Receiver Operating Characteristic (ROC). Each bar represents the cut-off values of each recombinant protein based indirect ELISA.

**Table 1 animals-14-00531-t001:** Detection of anti-*N. caninum* IgG in field cattle specimens from eight farms in Khon Kaen Province, Thailand using NcPrx2-, NcMIC4-, and NcSAG1-based indirect ELISA and IFAT (n = 247).

Farm	No. of Dairy Cattle	No. of Positive (%)			
		IFAT	NcPrx2	NcMIC4	NcSAG1
1	61	32 (52.4)	36 (59.0)	34 (55.7)	27 (44.2)
2	7	6 (85.7)	4 (57.1)	6 (85.7)	6 (85.7)
3	29	14 (48.2)	15 (51.7)	16 (55.1)	14 (48.2)
4	29	15 (51.7)	20 (68.9)	16 (55.1)	19 (65.5)
5	31	20 (64.5)	14 (45.1)	21 (67.7)	21 (67.7)
6	36	21 (58.3)	28 (77.7)	16 (44.4)	25 (69.4)
7	30	17 (56.6)	21 (70.0)	14 (46.6)	14 (46.6)
8	24	13 (54.1)	17 (70.8)	15 (62.5)	11 (45.8)
Total	247	138 (55.8)	155 (62.7)	138 (55.8)	137(55.4)

Abbreviations: NcPrx2, *N. caninum*-derived peroxiredoxin 2; NcMIC4, *N. caninum* microneme 4; NcSAG1, *N. caninum* surface antigen 1; ELISA, enzyme-linked immunosorbent assay; IFAT, indirect fluorescent antibody test.

**Table 2 animals-14-00531-t002:** Comparison of *N. caninum* IgG detection rate in field cattle serum samples using NcPrx2-, NcMIC4-, and NcSAG1-based indirect ELISA, with that of IFAT as the reference test (n = 247).

IFAT	NcPrx2		NcMIC4		NcSAG1	
	(+)	(−)	(+)	(−)	(+)	(−)
(+)	120	18	116	22	120	18
(−)	35	74	22	87	17	92
Total	155	92	138	109	137	110

Abbreviations: NcPrx2, *N. caninum*-derived peroxiredoxin 2; NcMIC4, *N. caninum* microneme 4; NcSAG1, *N. caninum* surface antigen 1; ELISA, enzyme-linked immunosorbent assay; IFAT, indirect fluorescent antibody test.

**Table 3 animals-14-00531-t003:** Sensitivity and specificity of indirect ELISAs using recombinant proteins for the detection of specific *N. caninum* IgG antibodies, with IFAT as the reference test.

Parameters	NcPrx2	NcMIC4	NcSAG1
Sensitivity (%)	87.0	84.1	88.4
Specificity (%)	67.0	78.9	80.7
Kappa value	0.558	0.64	0.713

Abbreviations: NcPrx2, *N. caninum*-derived peroxiredoxin 2; NcMIC4, *N. caninum* microneme 4; NcSAG1, *N. caninum* surface antigen 1; ELISA, enzyme-linked immunosorbent assay; IFAT, indirect fluorescent antibody test.

## Data Availability

All data relating to the present study are available in this manuscript and Appendix A.

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
