# Peer review of "Evaluation of Immunodiagnostic Performances of Neospora caninum Peroxiredoxin 2 (NcPrx2), Microneme 4 (NcMIC4), and Surface Antigen 1 (NcSAG1) Recombinant Proteins for Bovine Neosporosis"

_animals, 2024, doi:10.3390/ani14040531_

Round 1

Reviewer 1 Report

Comments and Suggestions for Authors

Dear authors,

The present document is related with an one of the main healthy problem for small ruminants around de world. Please, check the comments described into the file for improve the information and give to the readers more options to preventive and control the disease.

Reviewer 2 Report

Comments and Suggestions for Authors

The submitted manuscript is a study whose main aim is to present a comprehensive assessment of three N. caninum recombinant antigens: NcPrx2, NcMIC4, and NcSAG1, in terms of evaluating the serodiagnostic performance in an indirect ELISA format by comparing their efficacy with that of the Indirect Fluorescent Antibody Test, considered as an established reference method for diagnosing neosporosis in cattle.

The approach used to assess the diagnostic capability of the three different N. caninum antigens is a conventional one, thus no novelty is found in terms of technical innovation. Nonetheless, the results presented in the article are of interest to those researchers working on the bovine neosporosis clinical management/diagnostics area, as it has been shown that currently there is no available diagnostic gold standard for detecting antibodies to N. caninum in affected cattle.  

Overall, the manuscript is well written, the laboratory approach concerning the analytical methods utilized is adequate, and sufficient technical details are provided to replicate the work.

However. authors are invited to elaborate a little more, specifically in the discussion section, on an important issue concerning a substantial serological prevalence rate (55% - 62%) found in the bovine samples analyzed (274) from 8 cattle farms collected from randomly selected herds from the smallholder dairy farmers in Khon Kaen province, Thailand. This, particularly in the absence of endemic/epidemic cases of abortion, and as described in the materials and methods section of the manuscript the cattle were clinically healthy. Authors may refer to the following citation to better discuss and assert what kind of control practices could then be recommended in this type of epidemiological situation for the cattle premises in Thailand.

McAllister MM. Diagnosis and Control of Bovine Neosporosis. Vet Clin North Am Food Anim Pract. 2016;32(2):443-63. doi: 10.1016/j.cvfa.2016.01.012.

Thus, the study results could provide a resource for researchers and healthcare professionals addressing the impact of neosporosis in Thailand and other countries, by discussing the subject critically, identifying opportunity areas for diagnostic sero-surveillance and control approaches, pointing out existent research gaps in this particular disease.

In addition, minor details to be taken care off are as follows:

1.      Introduction

Line 66. Use “an economically important disease” instead of “serious”

Line 82. Can utilise “to detect” or “to identify” instead of “to diagnose” in the statement  “…and used to diagnose specific antibodies against N. caninum in cattle”

Line 85 Better use “may detect cross-reactivity with antibodies against other members of the phylum Apicomplexa”

Line 123. Check statement “The number of NcPrx2 and NcMIC4 proteins that have been investigated”. Should it rather be “The number of N. caninum proteins that have been investigated”?

2.      Materials and Methods

 Line 274. Change “fluorescein” to “fluorescence signal”

3.      Results

Lines 334-335. Delete paragraph “Abbreviations: NcPrx2, N. caninum-derived peroxiredoxin 2; NcMIC4, N. caninum microneme 4; NcSAG1, N. caninum surface antigen 1; ELISA, enzyme-linked immunosorbent assay; IFAT, indirect fluorescent antibody test”. 

4. Discussion

Line 407. Specify what “This substantial increase…” refers to. Should this paragraph be connected to that expressed in lines 381-386? If so, when the paragraph is changed, make sure the citation numbers are consecutively ordered

5. Conclusions

Line 456. Use “highlights” instead of “highlighting”

Line 460. Change “antigen marker for detecting N. caninum infection in cattle” to “…antigen marker for detecting antibodies to N. caninum in cattle”.

References

 Use abbreviated Journal name in references [3, 5, 11, 34, 36]

Comments on the Quality of English Language

 Minor editing of the English language required

Reviewer 3 Report

Comments and Suggestions for Authors

The manuscript “Evaluation of immunodiagnostic performances of Neospora caninum peroxiredoxin 2 (NcPrx2), microneme 4 (NcMIC4), and surface antigen 1 (NcSAG1) recombinant proteins for bovine neosporosis” by Udonsom et al. describes the validation of indirect ELISAs for the diagnosis of neosporosis in cattle, using recombinant Neospora caninum MIC4 and Prx2 proteins as antigens. The authors also include the evaluation of cattle antibody reactivity to SAG1, which was previously developed. After comparison of the sensitivity and specificity of each of the different iELISAs with the reference technique IFAT, the authors concluded that NcPrx2 and NcMIC4 could serve as potential diagnostic markers for detecting N. caninum infection in cattle.

However, although the sensitivity was quite reasonable, the specificity was low, particularly when NcPrx2 was used as antigen. These values are below the ones described for several commercial and experimental tests and thus limit the usefulness and relevance of these antigens for diagnosing neosporosis in cattle. Therefore, even though the studies are valid, this reviewer does not consider the authors conclusions to be supported by the results. It should be recognized that NcMIC4 and NcPrx2 antigens were tested, but have no advantage over others already described for the serological diagnosis of neosporosis.

There are some issues that should be addressed:

- No indication of animal ethics approval is provided. Only reference to a murine study where ethics approval is stated and to the samples obtained from the National Research Center for protozoan Diseases, Obihiro, Japan. No further information is provided for the field samples. Was sample collection exempt from ethics approval?

-Do the false negatives and false positives (in comparison with IFAT) in the different iELISAs match among antigens?

- NcMIC4 seems to be poorly purified (SDS-Page gel of Figure S2). Could this have contributed to decrease its specificity? 

-Studies from other authors, in which recombinant SAG1 was used for bovine neosporosis diagnosis by iELISA (e.g. Howe et al 2002, doi: 10.1128/CDLI.9.3.611-615.2002 and  Sinnott et al 2020, doi: 10.1016/j.vetpar.2020.109101), are not in the reference list.

- The authors acknowledge that immunoblotting should be considered as the reference test when developing new methods for N. caninum serodiagnosis, but they have not use this test in the present study, which is not in accordance with that statement. That would had soundness to the study, at least for the samples that were not concordant among tests.

- Discussion on differences in the prevalence of neosporosis in that geographical area is vague. The fact that "comprehensive information concerning the risk factors associated with N. caninum infection remained unavailable for this study" due to COVID-19 pandemics also limits comparisons. Even though access to herd information was restricted for a period, this limitation has since been overcome and records should be consulted to provide more information on the structure of the animal population and the dynamics of the infection.

Round 2

Reviewer 2 Report

Comments and Suggestions for Authors

The submitted manuscript presents the assessment of three N. caninum recombinant antigens: NcPrx2, NcMIC4, and NcSAG1, by evaluating the serodiagnostic performance in an indirect ELISA format and comparing their efficacy with that of the Indirect Fluorescent Antibody Test, an established reference method for diagnosing neosporosis in cattle.

 The approach used to assess the diagnostic capability of the three different N. caninum antigens is a conventional one, thus no novelty is found in terms of technical innovation. Nonetheless, the results presented in the article are of interest for those researchers working on the bovine neosporosis clinical management/diagnostics area, as it has been shown that currently there is not available a diagnostic gold standard for detecting antibodies to N. caninum in affected cattle.  

Overall, the manuscript is well written, the laboratory approach concerning the analytical methods utilized is adequate, and sufficient technical details are provided to replicate the work.